# EnvGen: Generating and Adapting Environments via LLMs for Training Embodied Agents

**Abhay Zala**[*]  **Jaemin Cho**[*]  **Han Lin**  **Jaehong Yoon**  **Mohit Bansal**
UNC Chapel Hill
{aszala, jmincho, hanlincs, jhyoon, mbansal}@cs.unc.edu
https://envgen-llm.github.io

## Abstract

Recent state-of-the-art approaches for embodied learning via interaction directly employ large language models (LLMs) as agents to determine the next steps in an environment. Due to their world knowledge and reasoning capabilities, LLM agents achieve stronger performance than previous smaller agents based on reinforcement learning (RL); however, frequently calling LLMs is slow and expensive. This begs an interesting question: *Instead of directly employing LLMs as embodied agents, can we use LLMs' reasoning capabilities to adaptively create training environments to help smaller embodied RL agents learn useful skills that they are weak at?* In this work, we propose **EnvGen**, a novel framework to address this question. First, we prompt an LLM to generate training environments that allow agents to quickly learn different tasks in parallel. Concretely, the LLM is given the task description and environment simulator objectives that the agents should learn and is then asked to generate a set of environment configurations (*e.g.*, different terrains, items initially given to agents, chances of finding certain objects, *etc.*). Next, we train a small RL agent in a mixture of the original and LLM-generated environments. Then, we enable the LLM to *continuously adapt* the generated environments to progressively improve the skills that the agent is weak at, by providing feedback to the LLM in the form of the agent's performance. We demonstrate the usefulness of EnvGen with comprehensive experiments in Crafter and Heist game environments. We find that a small RL agent trained with EnvGen can outperform SOTA methods, including a GPT-4 agent, and learns long-horizon tasks significantly faster. We also show that using an LLM to adapt environments dynamically outperforms curriculum learning approaches and how the LLM adapts training environments to help improve RL agents' weaker skills over time. Additionally, EnvGen is substantially more efficient as it only uses a small number of LLM calls (*e.g.*, 4 in total), whereas LLM agents require one or more LLM calls per step (resulting in thousands of LLM calls per episode). We also present detailed analyses of EnvGen's design choices.

## 1 Introduction

There has been growing interest in embodied AI, where agents learn through interactions with environments instead of static datasets (Ahn et al., 2022; Duan et al., 2022; Wang et al., 2023a; Yao et al., 2023; Driess et al., 2023). Open-world games such as Minecraft (Mojang Studios, 2009) and Crafter (Hafner, 2022) have been widely used as research environments for embodied agents, where the agents visually perceive their surroundings, traverse large terrains, and learn to unlock various *achievements* (*e.g.*, collecting resources, building tools, defeating monsters, *etc.*). Some achievements can be easily unlocked within a few steps, whereas others are more challenging as they only become accessible after the agent completes a series of prerequisite achievements, requiring hundreds of steps (*i.e.*, long-horizon tasks). As illustrated in Fig. 1 (a), traditional embodied agents are based on reinforcement

---

[*]equal contribution

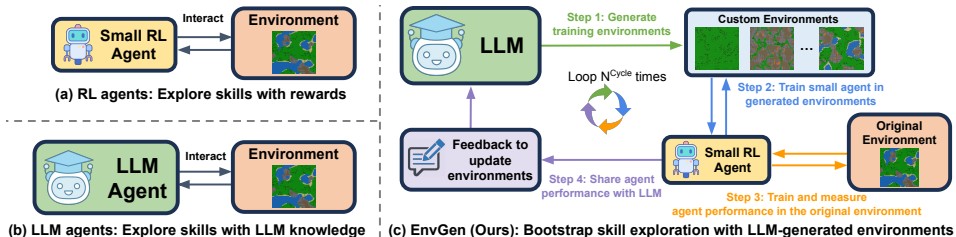

Figure 1: Comparison of different methods for creating embodied agents. Previous works commonly use **(a) small RL agents** or **(b) LLM agents** to explore skills. In **(c) EnvGen**, we train a small RL agent with diverse LLM-generated environments that train different skills in parallel and can be adapted via feedback to help the agents progressively improve skills that they are weaker at. Our method benefits from using the world knowledge from LLMs while maintaining efficient training through a lightweight RL agent.

learning (RL) (Hafner et al., 2020; 2021; 2023; Schulman et al., 2017; Burda et al., 2018; Hessel et al., 2018; Sekar et al., 2020; Moon et al., 2023). However, these RL agents usually struggle when learning such long-horizon tasks since the reward is sparsely given only after the correct execution of successive actions, and it is very expensive to automatically find many action sequences which lead to the reward (Aytar et al., 2018; Li et al., 2022a; Yuan et al., 2023), even after long pretraining with curiosity-driven intrinsic reward (Walker et al., 2023).

As large language models (LLMs) have shown remarkable progress in various tasks that require complex reasoning (Brown et al., 2020; OpenAI, 2023a; Touvron et al., 2023a;b; Chowdhery et al., 2023; Anil et al., 2023), recent works study implementing embodied agents based on LLMs. As illustrated in Fig. 1 (b), these methods leverage LLMs' world knowledge with chain-of-thought reasoning (Nye et al., 2021; Kojima et al., 2022; Wei et al., 2022) by creating action plans, giving feedback, and obtaining rewards throughout the episode (Yuan et al., 2023; Wang et al., 2023c; Wu et al., 2023; Wang et al., 2023a;d; Zhao et al., 2023; Du et al., 2023). While these LLM-based agents that verbalize their knowledge in reasoning steps have seen success in achieving better performance over previous approaches, iteratively calling LLMs throughout the episode is prohibitively slow and expensive (*e.g.*, SPRING (Wu et al., 2023) calls GPT-4 (OpenAI, 2023a) 9 times to take any action step, which results in $270 USD to complete an episode). Du et al. (2023) use LLMs to create rewards to train smaller agents, but the training is still costly, as it requires many interactions between the LLMs and student agents. This begs the question: *Instead of directly employing LLMs as embodied agents, can we use LLMs' reasoning capability to adaptively create training environments to help smaller embodied RL agents learn useful skills that they are weak at?*

To address this question, we propose **EnvGen**, a novel framework where an LLM adaptively generates training environments to teach smaller embodied RL agents. We aim to generate environments that can create various conditions (*e.g.*, have different terrains or some sub-goals are already achieved) so that agents can learn different skills in parallel and obtain more frequent rewards for challenging long-horizon tasks than in the original environment. As shown in Fig. 1 (c), EnvGen iterates over multiple training cycles, each consisting of the following four steps:

- Step 1: We generate configurations for custom training environments (*i.e.*, specifically created to train an RL agent on certain skills) by providing an LLM with a prompt including task description, controllable simulator settings, and simulator constraints (see Fig. 2 and Sec. 2 for details). Then we use the generated configurations to create different custom environments (*e.g.,* different terrains, items initially given to agents, and chance of finding certain objects) that can teach multiple skills in parallel.
- Step 2: We first train the RL agent in multiple LLM-generated environments (*i.e.,* LLM environments), so that it can learn different useful skills in parallel.
- Step 3: We then train the RL agent in the original environment to mitigate overfitting to the LLM environments. Then, we measure the current RL agent's performance in different tasks in the original environment to check which skills the agent is weak at.

- Step 4: We provide the RL agent's successes/failures in different tasks (from step 3) as feedback to the LLM, so that the LLM can adapt the custom training environments to focus on progressively improving the skills that the agent is weak at.

Note that EnvGen only requires a few LLM calls (*e.g.*, 4) for environment generation/updating during the entire RL agent training process, whereas other works based on LLM agents query an LLM once or multiple times every step (resulting in thousands of LLM calls for a single episode).

We study the usefulness of EnvGen in different game environments: Crafter (Hafner, 2022) and Heist (Cobbe et al., 2020). In the Crafter environment, a simple PPO-based (Schulman et al., 2017) lightweight (< 5M parameters) RL agent trained with our LLM-generated environments outperforms strong baselines including a GPT-4 based agent that queries an LLM multiple times at every step, and RL agents that use extensive pretraining (*e.g.*, 150M steps *vs.* less than 1M steps for us). When compared to just training longer in the original Crafter environment and curriculum learning approaches such as easy-to-hard and adversarial environments, an RL agent trained with EnvGen achieves significant improvements on the overall score and long-horizon tasks. In Heist, we also show that our LLM-generated environments can improve overall agent performance and training stability. We also show a qualitative study on how the LLM adapts training environments to help improve RL agents' weaker skills over time. Finally, we provide comprehensive analysis and ablation studies of the design choices of EnvGen, including dynamically updating LLM environments (*i.e.*, using adaptive environments) *vs.* curriculum learning methods, frequency of environment updates, EnvGen *vs.* longer training in the original environment, different LLMs for generating environments, the number of LLM-generated environments, and the mixture ratio between the original and LLM environment during training.

## 2 EnvGen: Generating and Adapting Environments via LLMs for Training Embodied Agents

We propose **EnvGen**, a novel framework where an LLM adaptively generates training environments to train smaller embodied RL agents, enabling them to accomplish various tasks within an environment, particularly long-horizon tasks. During the training process, the LLM is given feedback (in the form of the agent's performance) and can adaptively update the training environments to progressively focus on improving the tasks that the agent is weak at. In the following, we first explain why it is challenging to explore long-horizon tasks in open-world games (Sec. 2.1). Then we explain our method details, including how we generate environments and how agents are trained in EnvGen (Sec. 2.2).

### 2.1 Preliminary: Exploration is Hard for Long-Horizon Tasks

In the RL framework, agents explore various states along a trajectory and amplify policies based on the rewards received from those trajectories. However, exploration for long-horizon tasks is slow and computationally expensive, as rewards for such tasks are sparsely given only after a sequence of successful actions that often involve achieving multiple subgoals. For example, the goal in Crafter (Hafner, 2022) is to unlock 22 achievements, where some achievements can be unlocked quickly through several simple actions and others require long chains of prerequisites (*e.g.*, *collect iron* requires *make stone pickaxe*, which must be preceded by *collect stone*, ... *etc*.); see Sec. 3.1 for details. As shown in Hafner (2022), existing agents in Crafter spend most exploration steps learning low-level achievements but fail to unlock high-order achievements with many prerequisites.

### 2.2 EnvGen Method Details

We introduce **EnvGen**, where we train an embodied RL agent in multiple LLM-generated environments (we refer to these as 'LLM environments' in the paper) that progressively adapt to improve agent performance in multiple skills. The generated environments can provide various conditions (*e.g.*, different terrains, or some subgoals are already achieved)

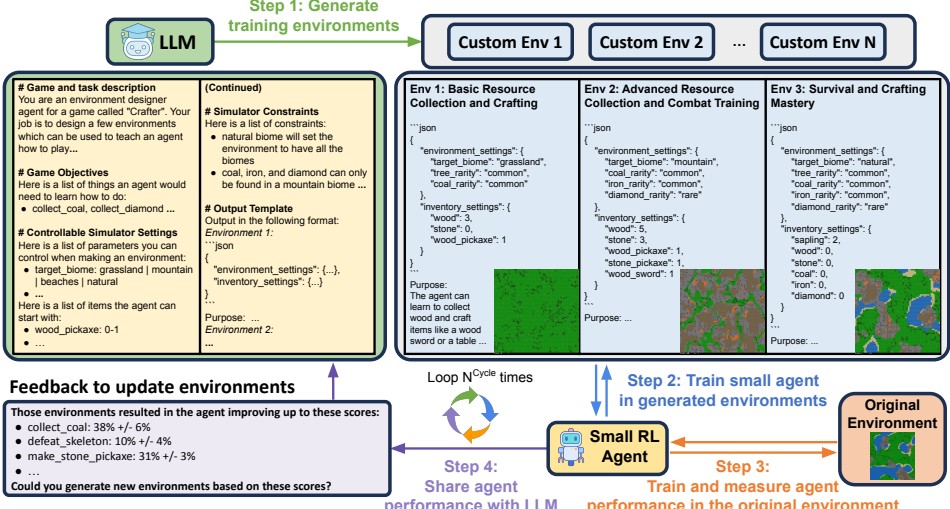

Figure 2: In EnvGen, we generate and adapt multiple training environments with an LLM to let the agent learn different skills effectively. EnvGen iterates over $N^{\text{Cycle}}$ cycles, each consisting of four steps (see Sec. 2.2).

so that agents can learn different skills in parallel and obtain more frequent rewards for long-horizon tasks. As shown in Fig. 2, EnvGen iterates $N^{\text{Cycle}}$ training cycles, each consisting of the following four steps:

**Step 1: Generate training environments with an LLM.** As illustrated in step 1 of Fig. 2, we use an LLM (*e.g.*, GPT-4 (OpenAI, 2023a)) to first generate $N^{\text{LLM-Env}}$ custom training environment configurations[1] that can cover various objectives and skills that are required in the original environment. The following describes the LLM input prompt components used to create environment configurations.

1. **Task description:** We provide a brief description of the environment and what the LLM should do (*e.g.*, "*generate a set of training environments...*").
2. **Game/simulator details:** We provide a list of objectives that need to be achieved in the environment (*e.g.*, "*collect coal, collect iron, etc.*" for Crafter); a list of which simulator settings can be controlled (*e.g.*, terrain, agent inventory); and a list of constraints/rules that the simulator has (*e.g.*, "*skeletons only spawn in mountains; ...*" for Crafter).
3. **Output environment configuration template:** We provide a blank output configuration template (*i.e.*, a JSON object where the environment settings are empty) to the LLM, and request it to fill in the values, creating $N^{\text{LLM-Env}}$ environment configurations. Along with filling the templates, we also ask the LLM to verbally explain the **purpose** for each environment (*e.g.*, what the environment would teach the agent); this would help users easily understand the environment generation process.
4. **Adaptation feedback based on the RL agent's performance**: We provide the LLM with the performance of the RL agent from the original environment (measured in step 3 and summarized in step 4), as feedback for adapting LLM environments to focus on skills that the RL agent is weak at. The feedback is given at the end of each cycle, so it is only provided to LLM from the second cycle onwards.

The obtained environment configurations are then rendered in the game's simulator. Fig. 2 presents the summary of input prompt and output environments from the GPT-4 model. We provide more prompt details in Appendix F.

---

[1]We find that N=4 works well; see Table 6 for details.

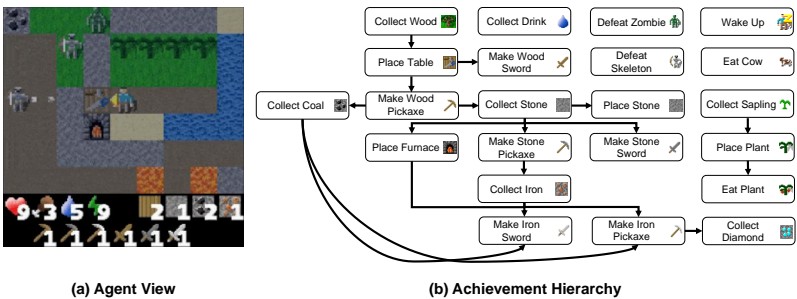

(a) Agent View  (b) Achievement Hierarchy

Figure 3: **(a): Crafter gameplay screenshot.** An agent explores a 2D world and completes 22 achievements. **(b): Crafter achievement hierarchy.** Some achievements can be completed right away; others require previous achievements to be unlocked first (*i.e.*, in a hierarchical order following the arrows).

**Step 2: Train a small RL agent in the LLM-generated environments.** As shown in step 2 of Fig. 2, we train the small RL agent in the LLM-generated environments. Concretely, we train the agent in the $N^{\text{LLM-Env}}$ LLM environments for $T^{\text{LLM-Env}}$ total steps in parallel.

**Step 3: Train and measure the RL agent's performance in the original environment.** It is important to note that the goal of EnvGen is to improve the RL agent's performance in the original environment, instead of the performance only in the LLM environments. To help the RL agent effectively adapt to the original environment and provide the LLM with the current agent's performance as feedback, we train the agent and measure its performance in the original environment, as shown in step 3 of Fig. 2. First, to mitigate the overfitting to LLM environments, we train the agent in the original environment for $T^{\text{Orig-Env}}$ steps.[2] Next, to find the skills that the RL agent needs to improve at, we test the agent in the original environment, without any parameter updates. Concretely, we measure individual success rates for each environment task (*e.g.*, Crafter achievements). The agent performance is summarized (in step 4) and is provided to LLM as feedback (in step 1) to adapt training environments in the next cycle. Moreover, importantly, to obtain a more calibrated estimation of agent performance, we calculate the average and variance of the task-specific scores by testing agents with multiple random seeds (*i.e.*, 12).

**Step 4: Send feedback to LLM to adapt environments (to focus on weak skills).** We provide the LLM with the agent's performance from the original environment (measured in step 3), as feedback for updating LLM environments. Concretely, we list the agent's average task-specific success rate in percentages along with one standard deviation (*e.g.*, "... *collect coal: 38% ± 6%, defeat skeleton: 10% ± 4% ...*"), as shown in step 4 of Fig. 2. In step 1 of the next cycle, the LLM can adaptively generate new environments (by using the agent's performance as feedback) to better help the RL agent learn the skills it is weak at (*e.g.*, defeat skeleton). EnvGen iterates this four-step training cycle $N^{\text{Cycle}}$ times.

## 3 Experimental Setup

In the following subsections, we present the benchmarks in which we evaluate EnvGen framework on (Sec. 3.1) and the agent architectures that we use for experiments (Sec. 3.2).

### 3.1 Evaluated Benchmarks and Training Details

**Crafter.** Crafter (Hafner, 2022) is an open-world 2D survival game focused on evaluating a broad range of agent capabilities (see Fig. 3). Crafter features 22 achievements that an agent can unlock during an episode of play. Some achievements can be unlocked in a few steps (*e.g.*, *collect wood*, *collect sapling*, *etc*.), but other achievements, such as *make iron pickaxe* or *collect diamond*, require many training/exploration steps and several prerequisite

---

[2]We find that $T^{\text{LLM-Env}} = T^{\text{Orig-Env}}$ works well; see Table 7 for details.

achievements to be unlocked (see Fig. 3 b). For example, to make an iron pickaxe, an agent must first collect enough wood to make a table and a wooden pickaxe, then go collect stone and return to the table (or collect more wood to make a new one) and then construct a stone pickaxe. Then the agent still needs to make a furnace, collect coal, and collect iron before the option to make the iron pickaxe is possible.

For EnvGen setup, we use $N^{\text{Cycle}} = 4$ training cycles during agent training (see Table 3 for ablation of having a different number of cycles). Each cycle uses 0.12M LLM-generated environment steps (*i.e.*, Crafter$^{\text{EnvGen}}$ steps, see step 2 in Fig. 2) and 0.12M Crafter steps (step 3 in Fig. 2) and then we train for 1M steps in Crafter. In total, we train for 1.96M steps ((0.12M + 0.12M) × 4 + 1M). Note that in order to maintain a fair score comparison to baselines, we do not count any achievement during our training cycle for score calculation since the training scores derived from LLM environments and the original environment are not directly comparable. Instead, we only take into account the achievements from the last 1M training steps in Crafter for the score calculation. See Appendix B for scoring details. We also experiment with giving the baseline model additional original environment steps to match the number of EnvGen steps (*i.e.*, an additional 0.96M steps) to ensure that EnvGen is not better simply because of more steps. We report the average performance with 30 runs (= 3 different initial LLM-generated Crafter$^{\text{EnvGen}}$ environments × 10 different random seeds).

**Heist.** Heist is part of the OpenAI Procgen (Cobbe et al., 2020) benchmark. In this environment, agents must successfully 'steal' the gem after navigating a maze and opening all locks. See more details in Appendix C.2.

### 3.2 Agent Architectures

**Our base RL agent.** For both Crafter and Heist, we test the EnvGen framework with a simple (CNN + linear layer) and lightweight (<5M) agent used in Moon et al. (2023), which is slightly modified from the agent architecture used in IMPALA (Espeholt et al., 2018). Following Moon et al. (2023), we train the agent with a PPO (Schulman et al., 2017) objective. At every step, the agent takes an RGB image (surroundings for Crafter, entire maze for Heist) as input and outputs the value estimates and policy (action probability). See Fig. 3 (a) for an agent visual input example. We provide additional model details in Appendix E.

**Baseline methods.** For Crafter, we compare our method to two groups of recent baselines – (1) methods that use frequent (*i.e.*, more than thousands of) LLM calls during training or inference: SPRING (Wu et al., 2023) (based on GPT-4) and ELLM (Du et al., 2023) (based on Codex (Chen et al., 2021)) and (2) methods that do not use an LLM: DreamerV3 (Hafner et al., 2023), MuZero + SPR (Walker et al., 2023), LSTM-SPCNN (Stanić et al., 2023), PPO (Schulman et al., 2017), and Achievement Distillation (AD) (Moon et al., 2023). For Heist, we compare against the PPO agent. For the PPO and AD agents, we follow the implementation of Moon et al. (2023). See Appendix E for the PPO/AD agent details.

## 4 Results and Analysis

We demonstrate the usefulness of the EnvGen method with comprehensive experiments and analysis. We first compare RL agents trained with EnvGen to different baseline methods on Crafter, an open-world game with 22 hierarchical achievements (Sec. 4.1). Next, we present a detailed analysis of the improvements that training with EnvGen environments can give RL agents on long-horizon tasks (Sec. 4.2). Then, we analyze how the LLM-based environment adaptation can help an RL agent progressively improve the skills that the agent is weak at (Sec. 4.3). Lastly, we present various additional analysis including experiments on Heist (a maze navigation game) and ablation studies on EnvGen design choices (Sec. 4.4 and also in the Appendix C).

### 4.1 Comparison with State-of-the-art Methods on Crafter Environment

**Small RL agent trained with EnvGen outperforms state-of-the-art baselines.** On the Crafter environment (described in Sec. 3.1), we compare a small PPO agent trained in

| Models | Description | # LLM calls | # Agent Params | Score (%) | Reward |
|---|---|---|---|---|---|
| Human* | | | | 50.5 ± 6.8 | 14.3 ± 2.3 |
| Random* | | | | 1.6 ± 0.0 | 2.1 ± 1.3 |
| ELLM* (Du et al., 2023) | 5M step PT in Crafter w/ Codex reward | 5M | 62M | - | 6.0 ± 0.4 |
| LSTM-SPCNN* (Stanić et al., 2023) | | | 135M | 11.7 ± 0.8 | 9.3 ± 0.2 |
| DreamerV3* (Hafner et al., 2023) | | | 201M | 14.8 ± 1.4 | 10.9 ± 0.5 |
| MuZero + SPR* (Walker et al., 2023) | 150M step PT in Crafter w/ RND reward | | 54M | 16.4 ± 1.5 | 12.7 ± 0.4 |
| SPRING* (Wu et al., 2023) | 9 queries to call GPT-4 per step | 2.7K[†] | Unknown | 27.3 ± 1.2 | 12.3 ± 0.7 |
| PPO (Moon et al., 2023) | | | 4M | 15.5 ± 0.6 | 10.5 ± 0.6 |
| PPO (Moon et al., 2023) | 0.96M step PT in Crafter | | 4M | 26.4 ± 2.1 | 12.1 ± 1.0 |
| AD* (Moon et al., 2023) | | | 9M | 21.8 ± 1.4 | 12.6 ± 0.3 |
| AD (Moon et al., 2023) | 0.96M step PT in Crafter | | 9M | 31.8 ± 0.7 | 13.3 ± 1.2 |
| PPO + EnvGen (Ours) | 0.96M step PT w/ Crafter[EnvGen] | 4 | 4M | 32.2 ± 0.6 | 12.6 ± 0.6 |
| AD + EnvGen (Ours) | 0.96M step PT w/ Crafter[EnvGen] | 4 | 9M | **35.3 ± 0.7** | 13.7 ± 0.8 |

Table 1: Comparison of different agents in the Crafter (Hafner, 2022) environment. Following previous works, we report the geometric mean of success rates across its 22 achievements and rewards for 1M Crafter steps. We experiment with EnvGen on two models, PPO and Achievement Distillation. *: scores from the Crafter Scoreboard (Hafner, 2022) and Moon et al. (2023). †: average number of LLM calls to run a single episode, according to SPRING (Wu et al., 2023). PT: Pretraining; AD: Achievement Distillation.

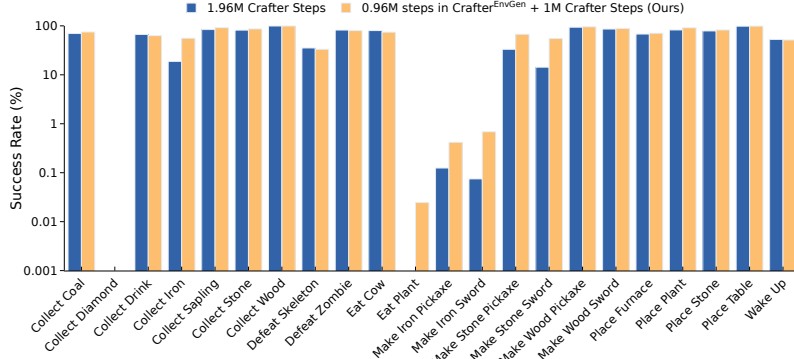

Figure 4: Success rates for all the Crafter achievements of two PPO agents (Moon et al., 2023) – (1) Baseline: trained in Crafter for 1.96M steps, and (2) Ours: trained in 0.96M steps in Crafter[EnvGen] and 1M in Crafter.

Crafter[EnvGen] (*i.e.*, Crafter environments generated with EnvGen) to state-of-the-art baseline methods. As shown in Table 1, we find that a small (4M parameters) PPO agent with EnvGen achieves an average score of 32.2% and significantly outperforms the baselines (and also in terms of the average reward). Note that some baseline agents have many more parameters or pretraining steps such as SPRING (GPT-4 agent; 27.3%), and MuZero + SPR (150M pretraining steps; 16.4%). Our method also only uses orders of magnitude fewer LLM calls (only 4) than works like SPRING (2.7K on average) and ELLM (5M), making it much cheaper/more efficient. EnvGen can also work with other RL agents such as Achievement Distillation (AD) (Moon et al., 2023) to achieve an even higher score (35.3%).

## 4.2 Detailed Achievement Analysis on Crafter Environment

Next, we analyze where EnvGen improves the overall score by checking individual achievement success rates in detail. For this, we compare the same PPO agent architecture trained with different setups: (1) an agent trained on Crafter for 1.96M steps and (2) an agent trained on Crafter[EnvGen] for 0.96M steps (0.24M steps × 4 training cycles, see Sec. 2.2) and then trained on Crafter for 1M steps. We measure the success rate (Fig. 4) of each achievement and unlocking speed (Fig. 5) of iron tools in the last 1M training steps.

**EnvGen helps RL agents to tackle challenging long-horizon achievements.** Fig. 4 shows that training in Crafter[EnvGen] improves scores of several achievements. Notably, training in Crafter[EnvGen] significantly improves the scores of long-horizon achievements (with many

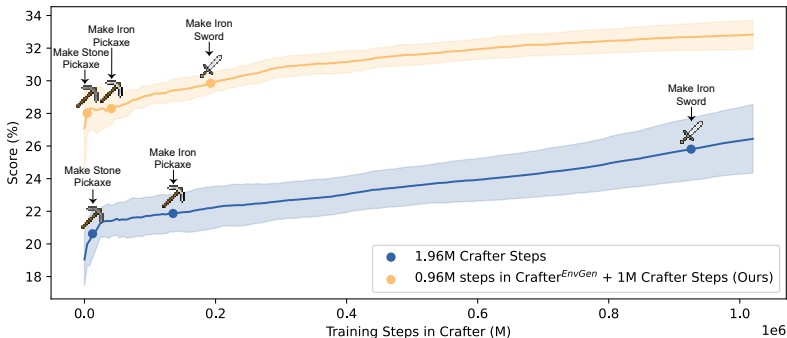

Figure 5: Unlock times (the first moment when the agent completed an achievement) for three long-horizon achievements ('*make stone pickaxe*', '*make iron pickaxe*', and '*make iron sword*') of two PPO agents (Moon et al., 2023) – (1) Baseline: trained in Crafter for 1.96M steps, and (2) Ours: trained for 0.96M steps in Crafter$^{EnvGen}$ and for 1M steps in Crafter. The plot shows the last 1M training steps out of 1.96M steps.

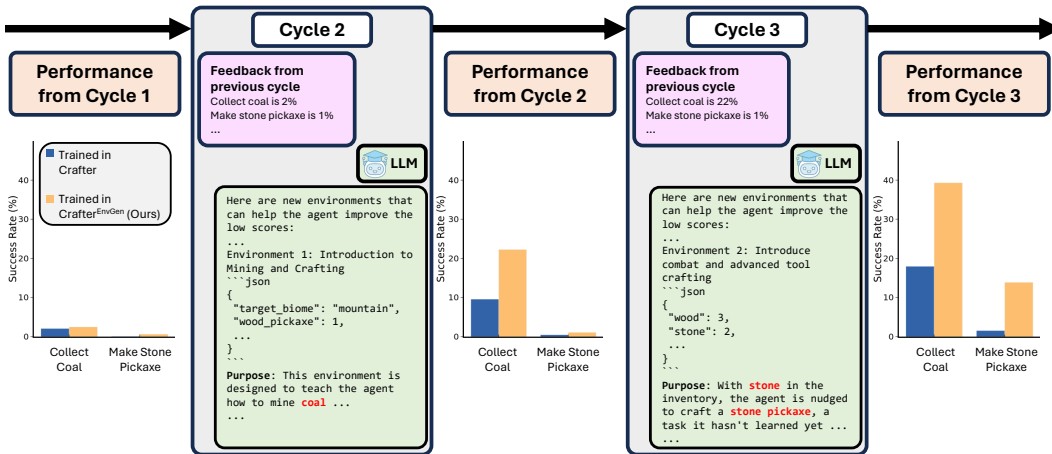

Figure 6: Adaptation of training environments based on agent performance over EnvGen cycles. At the end of each cycle, the RL agent's performance is given to the LLM as feedback (*e.g.*, '*Collect coal is 2%*'). The LLM uses the feedback to adaptively generate new environments that can help the agent progressively tackle skills it was previously weak at.

prerequisites; see Fig. 3) such as stone and iron tools. Fig. 5 shows that after unlocking the stone pickaxe, the RL agent trained in Crafter$^{EnvGen}$ is significantly faster in unlocking iron tools. In Appendix C.1, we also compare two AD agents, and show that Crafter$^{EnvGen}$ improves the success rate of the most challenging achievement – '*collect diamond*'.

## 4.3 Adaptation of Training Environments Helps the Agent Improve Weaker Skills

Fig. 6 shows how the LLM adaptively generates new training environments based on the intermediate performance of our PPO-based RL agent. In the intermediate performance plots, we compare the baseline agent trained only in Crafter and our RL agent trained in Crafter$^{EnvGen}$. In the cycle 2, given the feedback that the current RL agent is not good at collecting coal, the LLM generates an environment to help the agent focus on it, improving the agent's performance for the skill. Likewise, in the cycle 3, given the feedback that the agent is weak at making stone pickaxes, the LLM generates an environment to help the agent more easily craft the stone pickaxe, helping the agent improve it's score for the skill. Powered by the adaptive LLM environment generation of EnvGen, our agent learns to unlock these two achievements significantly faster than the baseline agent.

## 4.4 Additional Analysis and Ablation Studies

In the following, we provide analysis and ablation studies of EnvGen: dynamically updating LLM environments (*i.e.*, using adaptive environments) *vs.* curriculum learning methods, and different frequencies of environment updates. In Appendix C, we show comprehensive analysis and ablation studies of EnvGen method: EnvGen *vs.* longer training in the original environment, different LLMs for generating environments, the number of LLM environments, and the ratio of training steps in the LLM *vs.* original environments. We also include experiments on Heist environment (see Sec. 3.1) in Appendix C.2.

| Training Curriculum | Score (%) | Reward |
|---|---|---|
| Fixed (no curriculum) | $29.9 \pm 0.9$ | $12.6 \pm 0.8$ |
| Easy-to-Hard | $26.8 \pm 1.5$ | $12.7 \pm 0.7$ |
| Adversarial | $26.8 \pm 0.8$ | $12.2 \pm 0.7$ |
| Adaptive+Dynamic Environments (EnvGen) | $\mathbf{32.2 \pm 0.6}$ | $12.6 \pm 0.6$ |

Table 2: Comparison of RL agents trained in Crafter using no curriculum, an easy-to-hard curriculum, an adversarial curriculum, and our adaptive+dynamic environments. Agents are trained for 0.96M steps using the curriculum and then 1M steps in Crafter.

**Different environment curricula: fixed, easy-to-hard, adversarial *vs.* adaptive.** Table 2 shows that using LLM environments that are adaptively updated based on intermediate agent performance to improve weaker skills (last row) results in overall higher scoring agents than just using the initial LLM environments for the whole training (32.2% *vs.* 29.9%). These results indicate the effectiveness of the agent feedback and environment updating (step 4 described in Sec. 2).

Table 2 also compares an agent trained via EnvGen to the same agent trained with curriculum learning approaches such as an easy-to-hard curriculum, similar to Ammanabrolu et al. (2022) (*i.e.*, pre-defined training environment order based on environment difficulty) and adversarial curriculum, similar to Parker-Holder et al. (2022) (*i.e.*, updating to training environments that agent does worse in) in the Crafter environment. Detailed setups of both baseline approaches are in the appendix. The agent trained with EnvGen is able to achieve much higher performance (32.2% *vs.* 26.8% for both curricula) indicating the effectiveness EnvGen's approach of adaptively generating training environments to improve agent weak skills. The result indicates that creating more difficult environments does not necessarily help the agent learn new skills over time.

| Environment Update Frequency | # Training cycles $N^{\text{Cycle}}$ | Score (%) | Reward |
|---|---|---|---|
| Every 0.012M steps | 40 cycles | $30.8 \pm 0.7$ | $12.8 \pm 0.6$ |
| Every 0.06M steps | 8 cycles | $32.1 \pm 0.5$ | $12.7 \pm 0.8$ |
| Every 0.12M steps (default) | 4 cycles | $\mathbf{32.2 \pm 0.6}$ | $12.6 \pm 0.6$ |

Table 3: Different frequencies to update the environments (see Sec. 2 for details). Agents are trained with 0.96M steps in Crafter$^{\text{EnvGen}}$ and then 1M steps in Crafter.

**Frequency of LLM feedback / environment updates.** Table 3 shows that updating the LLM environments at every 0.12M steps results in the best agent performance. While increasing the cycles of environment feedback beyond 4 does not improve further, we find that updating environments with feedback always helps improve the RL agent's performance compared to training only with the original Crafter environment in Table 1 (26.4%) or the fixed LLM environment in Table 2 (29.9%).

## 5 Related Works

**LLMs as open-world game agents.** Recent works study using LLMs to create action plans (*i.e.*, a list of subgoals or skills to target) for embodied agents in open-world games like

Minecraft and Crafter (Hafner, 2022). Most of these methods require calling LLMs frequently (*e.g.*, at every step) for planning the next steps (Yuan et al., 2023; Wang et al., 2023c; Wu et al., 2023; Wang et al., 2023a;d; Zhao et al., 2023). Other methods, such as Li et al. (2024); Kwon et al. (2023); Ma et al. (2023); Du et al. (2023), have used LLMs to create/adjust rewards to train agents. Although these works show initial promising results leveraging the world knowledge of LLMs to tackle long-horizon tasks, iteratively calling LLMs throughout episodes is prohibitively slow and expensive (*e.g.*, running a single episode in the Crafter environment with SPRING (Wu et al., 2023) costs around $270 USD as they have 2.7K LLM calls on average). EnvGen only calls LLMs a few times (*e.g.*, 4 in total) to create training environments that focus on helping the RL agent progressively improve its weaker skills.

**Deep learning-based game/simulator content generation.** Procedural content generation (PCG) for games is about the automatic generation of levels, landscapes, items, rules, quests, or other types of game contents (Shaker et al., 2016). While traditional PCG methods are based on search/solver/rule/grammar-based methods, recent works apply deep learning methods such as GAN (Goodfellow et al., 2014) for PCG (Liu et al., 2021; Kumaran et al., 2020; Schubert et al., 2022). Several works have explored using LLMs to generate game content such as difficulty levels (Sudhakaran et al., 2023; Todd et al., 2023) and scenes/environments (Kumaran et al., 2023; Wang et al., 2023b; Afshar & Li, 2024). While these works aim to help developers create new game content, we aim to improve RL agent performance in the original environment. A line of work proposes unsupervised environment design (UED) that manipulates the difficulty level of environments to be more challenging to RL agents (Dennis et al., 2020; Jiang et al., 2021; Parker-Holder et al., 2022). While these works use a learned environment manipulator or evolutionary algorithms to maximize the 'regret' (the difference between the expected return of the current and optimal policies) in simple games such as MiniGrid (Chevalier-Boisvert et al., 2023), we use the world knowledge of LLMs to generate and adapt training environments that can improve weaker skills based on comprehensive skill-specific feedback from RL agents in open-world games with many challenging long-horizon tasks. To help agents generalize to unseen tasks in a text-based dialogue game, Ammanabrolu et al. (2022) augment new tasks with LMs and use a manually designed, fixed curriculum. Unlike this work, we adaptively generate training environments using LLMs' world knowledge and automatically learning a dynamic curriculum based on the RL agent's feedback, so as to improve the agent's weaker skills in open-world games with visual inputs. Beyond game content generation, several works visually augment vision-and-language navigation (VLN) simulators (*e.g.*, rendering environments with different styles) using image generation models (Li et al., 2022b; Wang et al., 2023e; Li & Bansal, 2023). Such works could complement our LLM environments (*e.g.*, augmenting our environments with diverse colors and textures).

# 6 Conclusion

We propose EnvGen, a novel framework to improve embodied RL agent performance by utilizing the world knowledge of LLMs to adaptively generate training environments. In EnvGen, we give an LLM a prompt describing a game/simulator and ask the LLM to generate the configurations to create new environments that can teach different skills. Next, we train an agent in the LLM-generated environments, give feedback to the LLM by testing the agent in the original environments, and then ask the LLM to update the environments to teach agents skills they are weaker at. In two challenging games, Crafter and Heist, we show that training in LLM-generated environments is significantly more effective than training longer in the original environments. We also show that using an LLM to adapt environments dynamically outperforms curriculum learning approaches and how the LLM adapts training environments to help improve RL agents' weaker skills over time. Moreover, a lightweight model ($< 5M$ parameters) trained with LLM-generated environments even outperforms an LLM agent with significantly fewer LLM calls. We hope our work can guide future works in leveraging LLMs for embodied agents.

## Acknowledgments

We thank Elias Stengel-Eskin and reviewers for thoughtful feedback. This work was supported by DARPA ECOLE Program No. HR00112390060, NSF-AI Engage Institute DRL-2112635, DARPA Machine Commonsense Grant N66001-19-2-4031, ARO Award W911NF2110220, ONR Grant N00014-23-1-2356, and a Bloomberg Data Science Ph.D. Fellowship. The views contained in this article are of authors and not of the funding agency.

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

# Appendix

In this appendix, we present additional related work (Appendix A), additional game environment details (Appendix B), additional experiment results (Appendix C), curriculum learning baseline method details (Appendix D), RL agent implementation details (Appendix E), additional LLM details (Appendix F), and limitations (Appendix G).

# A  Additional Related Works

**Reward designs in reinforcement learning.** Finding good action trajectories is critical in reinforcement learning (RL) (Sutton & Barto, 2018). While classic random exploration algorithms such as epsilon-greedy (Watkins, 1989) work well in simple settings such as multi-armed bandit, it is not the case for hard exploration problems where the environment gives very sparse rewards (Weng, 2020). A line of work studies how to augment the original (extrinsic) rewards from the environment with intrinsic rewards that encourage exploration (Bellemare et al., 2016; Burda et al., 2018). While such intrinsic rewards can help RL agents discover novel states and improve their knowledge about the environment, it often requires long pretraining and does not guarantee that the intrinsic reward can help the

target task. Another recent line of work studies using LLMs to adjust reward functions to help RL agents progressively learn certain tasks (Li et al., 2024; Kwon et al., 2023; Ma et al., 2023; Du et al., 2023). Instead of designing new rewards, in EnvGen, an LLM adaptively generates training environments that can help the RL agent learn multiple skills it is weak at with fewer training steps than in the original environment; reward design could be complementary to our method.

## B    Additional Game Environment Details

**Crafter.**    The score for Crafter is computed as the geometric mean of individual success rates of each achievement for each episode it is completed within 1M training steps: $S = exp(\frac{1}{22}\sum_{i=1}^{22} ln(1+s_i)) - 1$, where $s_i$ is the average success rate of the $i$th achievement across all episodes that occurred during training.

**Heist Environment.**    Heist is part of the OpenAI Procgen (Cobbe et al., 2020) benchmark. In this environment, agents must successfully 'steal' the gem after navigating a maze and opening all locks (see Fig. 7). The gem is behind three layers of color-coded locks, each requiring that the previous lock be unlocked first (*e.g.*, to unlock the green lock, the blue lock must first be unlocked). Following Moon et al. (2023), the final score is calculated as the average success of the agent in stealing the gem in 100 test episodes in 10 different seeds (*i.e.*, 1,000 runs in total). For agent training, we use a total of 5M steps in the LLM-generated environments (*i.e.*, 5M Heist$^{EnvGen}$ steps) and a total of 20M in the actual Heist environment. As the game only provides scores on the final objective ('*steal gem*') and the game is simple enough for the LLM-generated environments to cover all scenarios with one generation, we only use $N^{Cycle} = 1$ training cycle.

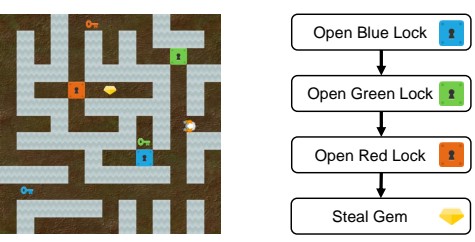

**(a) Agent View       (b) Achievement Hierarchy**

Figure 7: **(a)**: **Heist gameplay screenshot.** An agent aims to steal a gem (colored yellow), navigating a maze and colored opening locks. **(b)**: **Heist achievement hierarchy.** The agent can only reach the gem after successively unlocking all locks in order.

## C    Additional Experiment Results

### C.1    Design Choices, Ablations, and Other Agent Architectures

In the following, we show comprehensive design choice and ablation studies of EnvGen method: EnvGen *vs.* longer training in the original environment, different LLMs for generating environments, the number of LLM environments, and the ratio of training steps in the LLM environments to the original environment. Unless otherwise noted, we use the PPO-based agent (Moon et al., 2023) (described in Sec. 3.2) on the Crafter (Hafner, 2022) benchmark (described in Sec. 3.1) with 0.96M steps in Crafter$^{EnvGen}$ and average results for 30 runs (10 different seeds, 3 different initial environments).

**EnvGen *vs.* longer training in the original environment.**    Table 4 shows that when given an equivalent # of total training steps, the agents trained with Crafter$^{EnvGen}$ environments outperform the agents only trained with Crafter (*e.g.*, 22.3% *vs.* 21.1% for 1.24M total steps). Although the agent performances tend to improve with longer training steps in both settings,

| # Training Steps in Crafter$^{\text{EnvGen}}$ | # Training Steps in Crafter | Score (%) | Reward |
|---|---|---|---|
| *(Total 1.24M steps)* | | | |
| - | 1.24M | $21.1 \pm 2.3$ | $11.0 \pm 0.9$ |
| 0.12M | 1.12M | $\mathbf{22.3 \pm 1.5}$ | $11.6 \pm 0.8$ |
| *(Total 1.48M steps)* | | | |
| - | 1.48M | $21.9 \pm 2.1$ | $11.4 \pm 0.9$ |
| 0.24M | 1.24M | $\mathbf{27.9 \pm 1.2}$ | $12.4 \pm 0.7$ |
| *(Total 1.96M steps)* | | | |
| - | 1.96M | $26.4 \pm 2.1$ | $12.1 \pm 1.0$ |
| 0.48M | 1.48M | $\mathbf{32.2 \pm 0.6}$ | $12.6 \pm 0.6$ |

Table 4: RL agents trained in Crafter$^{\text{EnvGen}}$ environments *vs.* agents trained only in the Crafter environment. We calculate the scores based on the last 1M training steps in Crafter.

| LLM | Score (%) | Reward |
|---|---|---|
| Deepseek Coder 33B Instruct | $26.3 \pm 0.9$ | $12.1 \pm 0.8$ |
| GPT-3.5-Turbo | $21.5 \pm 2.8$ | $11.6 \pm 1.0$ |
| GPT-4-Turbo (default) | $\mathbf{29.9 \pm 0.9}$ | $12.6 \pm 0.8$ |

Table 5: Ablation of employing different LLMs to generate the environments. Agents are trained with 0.96M steps in Crafter$^{\text{EnvGen}}$ and 1M steps in the Crafter environment.

training with EnvGen shows stronger performance gains than only training longer in Crafter (*e.g.*, 32.2% *vs.* 26.4% for 1.96M total steps).

**Different LLMs to generate environments.** To figure out which LLM can generate more useful training environments, we experiment with three different LLMs (GPT-4-Turbo, GPT-3.5-Turbo (OpenAI, 2023b), and Deepseek Coder 33B Instruct (Guo et al., 2024)) and use $N^{\text{Cycle}} = 1$ (*i.e.*, fixed environment). Table 5 shows that environments generated by GPT-4-Turbo outperform that of other LLMs including GPT-3.5-Turbo and Deepseek Coder 33B Instruct. We see that GPT-3.5-Turbo performs the worst with only a score of 21.5%, while Deepseek 33B Instruct is able to get several points higher (26.3%) and GPT-4-Turbo, our default LLM, gets a few extra points (29.9%).

| # LLM environments | Score (%) | Reward |
|---|---|---|
| 1 | $30.8 \pm 0.5$ | $12.8 \pm 0.8$ |
| 2 | $29.1 \pm 0.6$ | $13.0 \pm 0.6$ |
| 4 (default) | $\mathbf{32.2 \pm 0.6}$ | $12.6 \pm 0.6$ |
| 8 | $31.0 \pm 0.8$ | $12.9 \pm 0.8$ |

Table 6: Different number of LLM environments being generated by the LLM per cycle. Agents are trained with 0.96M steps in Crafter$^{\text{EnvGen}}$ and 1M steps in the real Crafter environment.

**Number of LLM environments.** Table 6 shows that changing the number of environments generated by the LLM at each cycle (*i.e.*, 1, 2, 4, and 8) can slightly affect agent performance. While training with four environments produces the highest result, training with environments generated with any of the tested configurations improves performance over training only with the original Crafter environment (26.4%).

**Ratio of training steps: LLM environments *vs.* original environment.** As mentioned in Sec. 2.2, in EnvGen, we train the RL agent in LLM environments (step 2) and then in the original environment (step 3) to mitigate the agent from overfitting to the LLM environments. We experiment with different ratios of training steps in LLM environments (*i.e.*, Crafter$^{\text{EnvGen}}$) compared to training steps in the original Crafter environment (*e.g.*, 2:1

| Ratio of Training Steps in Crafter$^{EnvGen}$ : Crafter | Score (%) | Reward |
|---|---|---|
| 5:1 | $30.3 \pm 0.6$ | $12.3 \pm 0.9$ |
| 2:1 | $30.1 \pm 1.1$ | $12.8 \pm 0.7$ |
| 1:1 (default) | $\mathbf{32.2 \pm 0.6}$ | $12.6 \pm 0.6$ |

Table 7: Different ratios of training steps in LLM-generated environments (Crafter$^{EnvGen}$) compared to training steps in the original Crafter environment (*e.g.*, 2:1 indicates that for every two training steps in Crafter$^{EnvGen}$, the RL agent gets one training step in Crafter). We keep the total number of training steps constant at 1.96M.

indicates that for every two training steps in Crafter$^{EnvGen}$, the RL agent gets one training step in Crafter). As shown in Table 7, while different ratios do not result in big differences, the default 1:1 ratio provides the highest scores.

**Can simulators always understand LLM-generated environment configurations?** We tested and analyzed the generated environments used in paper experiments (109 total) and found that the LLM (GPT-4-Turbo) did not generate any environments beyond what the Crafter or Heist simulators could handle. While we do not find any such case, even if the LLM generates an invalid setup, we constrain all LLM-generated settings in post-processing to be within simulator capabilities to ensure no accidental errors in the simulator or environment generation.

**Environment parameter naming: obscure *vs.* original.** To determine whether or not the LLM in EnvGen is leveraging world knowledge when generating environments we conduct an analysis experiment. We replace environment parameter names from the original ones with obscure names (see Fig. 9), in order to remove the use of prior knowledge/expectation of how each parameter is useful to which skills. We find that the performance decreases, from $32.2 \pm 0.6 \rightarrow$ to $28.5 \pm 0.6$, indicating that the world knowledge/prior knowledge of the LLM is beneficial in helping the LLM generate more suitable environments.

**Achievement Distillation + EnvGen.** As mentioned in the Sec. 4.1, we also experiment using EnvGen with the Achievement Distillation (AD) (Moon et al., 2023) agent. As shown in Fig. 8, similar to our results on the PPO-based agent, we find that by applying EnvGen, there is performance gain in long-horizon tasks like making iron tools and collecting diamonds.

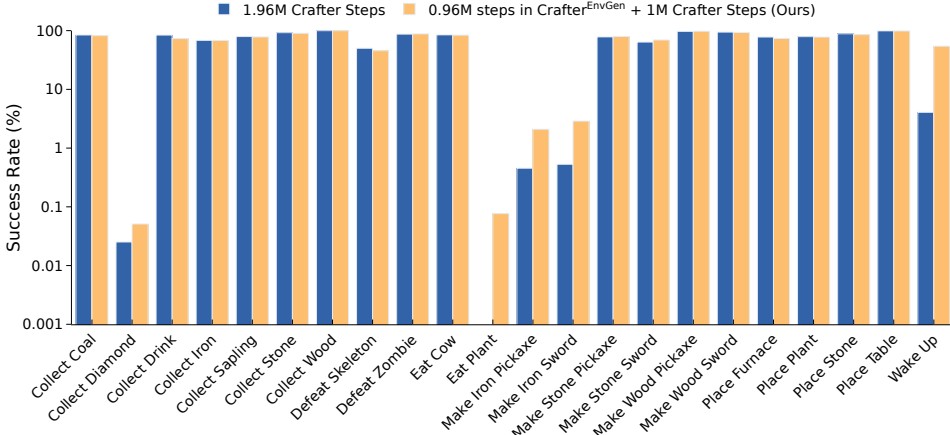

Figure 8: Success rates for all Crafter achievements of two Achievement Distillation (AD) agents (Moon et al., 2023): (1) Baseline: trained in Crafter for 1.96M steps, and (2) Ours: trained in Crafter$^{EnvGen}$ for 0.96M steps and Crafter for 1M steps.

| Model | # Training Steps in Heist$^{\text{EnvGen}}$ | # Training Steps in Heist | Score (%) | Reward |
|---|---|---|---|---|
| PPO | - | 25M | $25.9 \pm 13.2$ | $4.1 \pm 1.8$ |
| PPO + EnvGen (Ours) | 5M | 20M | $\mathbf{37.7 \pm 7.50}$ | $5.5 \pm 0.9$ |

Table 8: Evaluation results on Heist. Scores are computed as the average success rate over 100 test episodes over 10 different seeds.

## C.2 Evaluation on Heist Environment

**EnvGen can generalize to Heist.**  We also evaluate the effectiveness of EnvGen framework with Heist. We compare the PPO-based agent trained with and without EnvGen (*i.e.*, Heist$^{\text{EnvGen}}$ environments). Table 8 shows that training an agent with Heist$^{\text{EnvGen}}$ environments is effective in improving performance by increasing average scores (25.9% $\rightarrow$ 37.7%) and rewards (4.1% $\rightarrow$ 5.5%), while also stabilizing training by reducing the score variance (*i.e.*, standard deviation goes down 13.2% $\rightarrow$ 7.5%).

# D Curriculum Learning Baseline Details

In the following, we describe the two baseline implementation details described in Table 2: easy-to-hard and adversarial curricula.

**Easy-to-hard curriculum.**  Similar to Ammanabrolu et al. (2022), we create an easy-to-hard curriculum. An easy-to-hard curriculum has a pre-defined order of training environments. The agent first trains in "easy" environments and then by the end of the training process will be training in the "hard" environments. To do this, we first ask the LLM to generate a set of 16 random environments and train a validation agent (an agent only for the purpose of testing an environment difficulty; not used during final agent training) on each environment. Then the performance of the validation agent is measured and the environments are sorted from easiest to hardest (*i.e.*, from environments that resulted in higher agent scores to lower agent scores). Then we train an agent on the top four easier environments first and for every 0.24M steps we replace the training environments with the next four environments in the sorted set (*i.e.*, easy-to-hard curriculum).

**Adversarial curriculum.**  Similar to Parker-Holder et al. (2022), we create an adversarial curriculum. The adversarial curriculum approach involves updating the agent's training environments to ones where it has struggled. To do this, we first generate a set of 16 random environments and train a validation agent on each environment. Then, we measure and sort the environments by difficulty (*i.e.*, sorted from lowest to highest based on the validation agent's score). We take the top four hardest environments and train the final agent on these. Then, generate a new set of environments and test the current agent on this set, again sorting by difficulty. Then again, we take the top four hardest environments and resume training on these. This process then repeats four times (every 0.24M steps) creating an adversarial curriculum.

# E RL Agent Implementation Details

**PPO agent.**  We use the PPO-based (Schulman et al., 2017) agent used in (Moon et al., 2023), which modifies the default ResNet (He et al., 2016) architecture in IMPALA (Espeholt et al., 2018) by increasing channel size and hidden size and adding a layer normalization Ba et al. (2016) before each linear/convolutional layer. We slightly modify this architecture further to place the layer norm after the final linear layer instead of before. Hyperparameters for this model are shown in Table 9.

**Achievement distillation (AD) agent.**  Moon et al. (2023) builds upon its PPO-based agent model and adds auxiliary training steps after the PPO policy updates. Their auxiliary

**Before:**
Here is a list of parameters you can control when making an environment:
target_biome: grassland — mountain — beaches — natural
coal_rarity: very common — common — rare
iron_rarity: very common — common — rare
diamond_rarity: very common — common — rare
tree_rarity: very common — common — rare

Here is a list of items the agent can start with:
sapling: 0-9
wood: 0-9
stone: 0-9
coal: 0-9
iron: 0-9
diamond: 0-9
wood_pickaxe: 0-1
stone_pickaxe: 0-1
iron_pickaxe: 0-1
wood_sword: 0-1
stone_sword: 0-1
iron_sword: 0-1

**After:**
Here is a list of parameters you can control when making an environment:
parameter1: optionA — optionB — optionC — optionD
parameter2: optionE — optionF — optionG
parameter3: optionE — optionF — optionG
parameter4: optionE — optionF — optionG
parameter5: optionE — optionF — optionG

Here is a list of items the agent can start with:
item1: 0-9
item2: 0-9
item3: 0-9
item4: 0-9
item5: 0-9
item6: 0-9
item7: 0-1
item8: 0-1
item9: 0-1
item10: 0-1
item11: 0-1
item12: 0-1

Figure 9: LLM prompt template for environment generation before and after parameter and item names are replaced with obscure names.

training consists of two parts: (1) intra-trajectory achievement prediction and (2) cross-trajectory achievement matching. (1) Intra-trajectory achievement prediction maximizes the similarity between state-action pairs and the corresponding next achievement that needs to be unlocked in the achievement hierarchy within an episode. (2) Cross-trajectory achievement matching maximizes the similarity between achievements across episodes. Hyperparameters for this model are shown in Table 10.

## F    Additional LLM Details

**Prompt Template.**    In Fig. 10 (a), we show the LLM prompt template that is used to generate environments. The contents of the prompt can vary slightly between different environments/games though generally remain the same. In Fig. 10 (b), we show the additional prompt template that is used during the feedback step (step 4 in Sec. 2.2). At

| Hyperparameter | Value |
| --- | --- |
| Discount factor | 0.95 |
| GAE smoothing parameter | 0.65 |
| # timesteps per rollout | 4096 |
| # epochs per rollout | 3 |
| # mini-batches per epoch | 8 |
| Entropy bonus | 0.01 |
| PPO clip range | 0.2 |
| Reward normalization | No |
| EWMA decay rate | 0.99 |
| Learning rate | 3e-4 |
| Max grad norm | 0.5 |
| Value function coefficient | 0.5 |

Table 9: PPO agent hyperparameters. Hyperparameters are following Moon et al. (2023).

| Hyperparameter | Value |
| --- | --- |
| Policy regularizer coefficient | 1.0 |
| Value regularizer coefficient | 1.0 |
| Entropic regularizer coefficient | 0.05 |
| # policy phases per auxiliary phase | 8 |
| # epochs per auxiliary phase | 6 |

Table 10: Achievement Distillation hyperparameters. Hyperparameters are following Moon et al. (2023).

each feedback cycle iteration, the additional prompt is concatenated to the previous LLM output (*i.e.*, maintaining a chat history).

**API Cost.** As we use GPT-4-Turbo (1106-preview version) the API cost is $10.00 per 1M tokens and $30.00 per 1M tokens. The initial environment generation cost is $0.03 and then each iteration of the feedback cycle adds $0.04. Once the model is trained via EnvGen it no longer requires any LLM calls for inference or further training on the original environment. Works like SPRING (Wu et al., 2023) require $270 USD and several thousand LLM calls per episode, which is much more expensive than our work.

## G  Limitations

EnvGen relies on strong LLMs (*e.g.*, GPT-4). But note that one of the main motivations of EnvGen is to more efficiently use LLMs to help train embodied agents, and as such EnvGen requires very few LLM calls (*e.g.*, 4 calls), which only costs less than $1 USD during the entire training. We hope that advances in quantization/distillation and open-source models will make strong LLMs more accessible.

EnvGen also requires that the environment simulators can (or be easily edited to) accept configurations in standard formats (*e.g.*, JSON, CSV, YAML, TOML *etc.*), and the LLM can correctly generate configurations in such formats. Note that such text configuration formats are widely used for managing game simulators. In addition, many games have open-source community-driven efforts that provide high-level configuration documentation and settings, such as Minecraft wrappers (Guss et al., 2019; Fan et al., 2022) and Starcraft wrappers (Vinyals et al., 2017; DI-star Contributors, 2021).

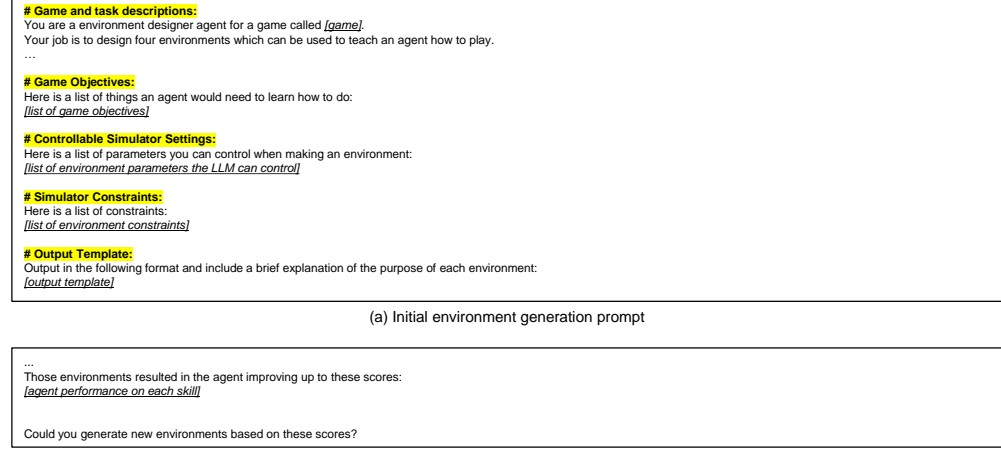

(a) Initial environment generation prompt

(b) Feedback cycle prompt

Figure 10: The prompts that are given to the LLM to generate environments in step 1 of Sec. 2.2.

