# OpenReview forum: "EnvGen: Generating and Adapting Environments via LLMs for Training Embodied Agents"
_colmweb.org/COLM/2024/Conference — COLM_

### Official Review · Reviewer_y9sf · 2024-05-05

**Rating:** 7
**Confidence:** 3
**Ethics Flag:** 1

**Summary:**

The paper proposed EnvGen, and LLM based small RL agent training environment dataset generation framework. The method prompts an LLM with game environment task descriptions and objectives, and obtains a more diverse gaming environment configurations for small RL model training. The method also evaluates the performance of the models and provides feedback to prompt LLM to generate game configurations that are more tailored to the RL models in order to improve their performance on the weaker tasks.

The paper conducted experiments on two gaming environment: Crafter and Heist, and compared to both LLM based zero shot agents and trained RL based agent. The paper demonstrated that the proposed method only requires a few LLM API calls and can improve RL agents' performance.

**Reasons To Accept:**

1. The paper is well written. It clearly explained the proposed method, conducted thorough experiments and demonstrated the benefits of the method
2. Instead of applying LLMs for a specific task, the proposed method used LLM to generate data for more efficient smaller RL model training
3. The feedback loop allows the method to generate customized curriculum for the learning agent.

**Reasons To Reject:**

1. To use LLMs to generate training data, it is under the assumption that these generated data is appropriate and close to the real environment. There wasn't any evaluation on the quality of the generated data
2. To demonstrate the benefits of the feedback loop and curriculum learning, it would be more convincing to have some more comprehensive quantitative results other than one example (w/"collect coal", "make stone pickaxe" ) in Figure 6.

---

> ### Author Rebuttal · Authors · 2024-05-30
>
> Thank you for your valuable feedback.  Below, we address your concerns with further clarifications and experiments.
>
> **W1. Quality of LLM-generated data:**
> From our testing and analysis of generated environments used in paper experiments, the LLM did not generate any environments beyond what the simulator could handle. While we do not find any such case, even if the LLM generates an invalid setup, we constrain all LLM-generated settings in post-processing to be within simulator capabilities to ensure no accidental errors in the simulator or environment generation.
>
> Additionally, we would like to clarify that the goal of the LLM is to generate environments that are not necessarily close to the real environment, but rather to target weak skills specifically. While in some cases, this may result in environments that are similar to the real one, it is not a requirement for good performance. As we show in Table 1 and Figures 4, 5, and 6, compared to a baseline result that is just training in the real environment, our method has significant performance gain, showing the effectiveness of the LLM-generated environments despite them not always resembling the real environment.
>
>
> **W2. Quantitative results for feedback loop:**
> We bring your attention to Table 3, which shows that if we remove the feedback loop/adaptive curriculum learning, the performance drops a few points (32.2 $\rightarrow$ 29.9).
>
> Then, we also develop a static, easy-to-hard curriculum strategy as follows:
> - (1) We generate a set of environments and train a test agent on each environment
> - (2) The performance of the test agent is measured
> - (3) The environments are sorted from easiest to hardest (e.g., environments that resulted in the higher agent scores to lower agent scores).
>
> Then, we train the baseline agent on the easier environments first and progressively move to the harder ones (e.g., easy-to-hard curriculum).
>
> |  | Score (%) | Reward |
> |---|---|---|
> | Static Easy-to-Hard Curriculum | 26.8 +/- 1.5 | 12.7 +/- 0.7 |
> | EnvGen Adaptive Curriculum (ours) | **32.2 +/- 0.6** | 12.6 +/- 0.6 |
>
> We see that our feedback loop/adaptive curriculum can also outperform this baseline by several points, again highlighting the effectiveness of our method.

---

> > ### Comment · Reviewer_y9sf · 2024-06-05
> > **Reviewer Response to Rebuttal**
> >
> > Thanks for your clarifications.

---

### Official Review · Reviewer_oNhz · 2024-05-15

**Rating:** 6
**Confidence:** 3
**Ethics Flag:** 1

**Summary:**

This paper proposes a new framework, EnvGen, that uses large language models (LLMs) to generate training environments for embodied agents. Instead of (1) directly employing LLMs as embodied agents, or (2) directly training RL agents in the environment, the authors propose to use LLM reasoning capabilities to adaptively create training environments to help smaller embodied RL agents learn useful skills that they are weak at.

The framework consists of 4 main steps:
1. Ues LLM to generate training environments that allow agents to quickly learn different tasks in parallel.
2. Train small RL agents in the generated environments and potentially learn different skills.
3. Train and measure agent performance in the original environment
4. Use the agent's performance to provide feedback to the LLM, and ask LLm to adapt the generated environments to improve the agent's performance.

The experiment results show that with this EnvGen framework, a small RL agent can outperform large LM and RL agents and learn long-horizon tasks significantly faster.

**My general review on the paper:**

I think this paper is presenting a new perspective on **utlizing** LLMs in embodied learning. Instead of training the LLM or calling LLMs at each step, the authors propose to use LLMs as part of a curriculum learning framework (even though the paper does not explicitly mention this). The LLM is essentially used to generate training environments that help the RL agent learn faster and design curriculum that is tailored to the agent's weaknesses.

The experiments demosntrate the effectiveness of the proposed framework, and I think they are comprehensive as it compares with non-LLM and LLM-based methods. Analsysis also show insights on how the method can help tha agent learn more efficiently.

There are a few downsides of the paper too. First or all, I think the paper is very closely related to **curriculum learning**, but the authors do not mention this in the paper, nor do they compare with existing curriculum learning methods. Second, the paper does not demosntrate the **upper bound of performance** of this method (e.g., using a larger RL agent or train with much more steps). Currently, the paper only emphasizes the efficiency of the method, but it is not clear how well the method can scale to more complex tasks and larger agents.

**Questions To Authors:**

1. Why can the LLM generate training environments that help the agent learn faster? What is the intuition behind this? Is it because the LLM has prior knowledge about the environment or is the LLM only inferrencing from the task description?

2. In page 2, you have "Step 2: We train the RL agent ...", but then "Step 3: We first train the RL agent". The word "first" seems a little confusing since you first trained in the LLm-generated environment in step 2.

3. In Figure 4, you compare (1) Baseline: trained in Crafter for **1.96M** steps, and (2) Ours: trained in **0.96M** steps in Crafter-EnvGen and **1M** in Crafter. Why is this comparison useful? Is training in Crafter-EnvGen more efficient than training in the original Crafter environment?

4. In Section 5, "To help agents generalize to unseen tasks in a text-based dialogue game, Ammanabrolu et al. (2022) augment new tasks with LMs and use a manually designed, fixed curriculum." Do you think the proposed method can be applied to this setting? If so, how would your method compare with the existing method?

**Reasons To Accept:**

1. The paper presents a novel framework that uses LLMs to generate training environments for embodied agents.
2. The experiments show that the proposed framework can outperform existing methods and help agents learn faster.
3. This is a new perspective on using LLMs in embodied learning, and the paper is well-written and easy to follow.

**Reasons To Reject:**

1. A lack of comparison with existing curriculum learning methods (both fixed curriculum and adaptive curriculum).
2. Not enough comparison with other content generation methods. A lot of work are menthioned in the related work section "Deep learning-based game/simulator content generation", but it seems none of them are compared with the proposed method.

---

> ### Author Rebuttal · Authors · 2024-05-30
>
> Thank you for your valuable feedback. Below we address your concerns with further clarifications and experiments.
>
> |  | Score (%) | Reward |
> |-|-|-|
> |(1) Static Easy-to-Hard | 26.8 +/- 1.5 | 12.7 +/- 0.7 |
> |(2) Adversarial | 26.8 +/- 0.8 | 12.2 +/- 0.7  |
> |(3) EnvGen Adaptive w/ Obscured Prompt | 28.5 +/- 0.6 | 12.6 +/- 0.9 |
> |(4) EnvGen Adaptive (ours) | **32.2 +/- 0.6** | 12.6 +/- 0.6 |
>
> **Table A:** Agents trained with different curricula.
>
> **W1/W2/Q4. Comparison to additional curriculum baselines:**
> We bring your attention to Table 3 in the main paper which already shows that our adaptive curriculum outperforms no curriculum (i.e., fixed). Even so, following your suggestions, we also experiment with two curriculum baselines from related works:
> - (1) Static easy-to-hard - Ammanabrolu et al. (2022)
> - (2) Adversarial curriculum - Parker-Holder et al. (2022)
>
> Table A shows that EnvGen’s adaptive curriculum outperforms the existing curriculum baselines by a large margin. Thanks for the suggestion. We will add the results (with more detailed experiment setup descriptions) to the camera-ready version.
>
> **Q1. Why are LLM environments helpful?:**
> As described in Sec 2, in EnvGen, an LLM uses agent performance as feedback to adaptive training environments. Following your suggestion, to see how much world knowledge of LLM is helpful, we additionally experiment with replacing environment parameter names from original ones with obscure names (e.g., “biome” into “parameter1”, “grassland” into “optionA”), to remove the prior knowledge/expectation of how each parameter is useful to which skill.
>
> Table A shows that obscuring parameter names decreases performance, but still outperforms other curriculum baselines, indicating that:
> - LLMs' world knowledge is useful in helping them find which environment parameters are more relevant to improving specific agent skills,
> - Dynamically adapting environments can be more effective than static/adversarial curricula.
>
> **Q2. Method description clarification:**
> Thanks for the suggestion, we will reword those sentences to be clearer.
>
> **Q3. Baseline purpose:**
> The goal of showing this comparison was to show that training an agent with the EnvGen framework allows the agent to get a significant score increase over an agent trained without EnvGen with the exact same number of steps (i.e., 1.96M). As shown in Figure 5, we also show that using EnvGen allows the agent to learn skills much quicker than when just training on Crafter.

---

### Official Review · Reviewer_Nrar · 2024-05-18

**Rating:** 6
**Confidence:** 3
**Ethics Flag:** 1

**Summary:**

The paper EnvGen, which uses LLM to construct custom training environment configurations for RL agent training in a iterative process. Experiments on Crafter and Heist game environments shows this method allows training RL agent more efficiently and achieve higher performance, while also outperforming baselines which directly promps a LM.

**Questions To Authors:**

- Minor: Figure 5 is a bit confusing to me. While the blue curve is supposed to show how the PPO agent trained on Crafter  improves over its 1.96M steps, the x-axis only has 1M steps. It would be good if you can elaborate that in the caption

**Reasons To Accept:**

- The method provides an novel, interesting perspective in using LMs for environment generation to assist RL agent training, and shows good empirical results. It can be a helpful addition to prior approaches like using LM as reward, using LM as agent, ...

**Reasons To Reject:**

- It is unclear if such methods can be applied in broader, more complex scenarios. As noted by the author, “EnvGen also requires that the environment simulators can [skipped] accept configurations in standard formats (e.g., JSON, CSV, YAML, TOML, etc.),". This is in fact a very strong assumption. Games like Minecraft, modern video games like GTA, or environments in other domains like robot training, might have too many degrees of freedom where such high-level configurations do not exist or require extensive human engineering to make.

---

> ### Author Rebuttal · Authors · 2024-05-30
>
> Thank you for your valuable feedback. Below we address your concerns with further clarifications.
>
> **W1. Adding configurations to complex games:**
> While it takes initial human engineering to build a wrapper to environment simulators, we would like to point out that (1) it is very common to store environment variables in standard configuration formats during the development process and (2) the amount of effort needed to build such wrapper primarily depends on the how much documentation of target simulator is available. For example, it is much easier to build wrappers for Minecraft than GTA, as there is already a community-driven effort of documentation and open-source projects, such as MineRL/MineDojo (which already support high-level configurations).
>
>
> **Q1. Clarification on Fig 5:**
> As described in Sec 3.1 and Appendix B, the Crafter score is calculated during 1M training steps in the Crafter environment. So, in our experiments, when we compare two agents trained for a total of 1.96M steps,
> - (1) the baseline agent trained only on Carfter for 1.96M steps, and
> - (2) our agent first trained with the mixture of Crafter and LLM-generated environments for 0.96M steps (= (0.12M + 0.12M) × 4) and then trained only on Crafter for 1M steps.
>
> We compare the scores/achievements during the last 1M steps in Crafter. The Fig. 5 shows the achievements unlocked during the 1M steps in Crafter. We will add clarification in the caption.

---

> > ### Comment · Reviewer_Nrar · 2024-06-06
> >
> > Thank you for your response! I am keeping my scores.

---

### Decision · Program_Chairs · 2024-07-10

**Decision:**

Accept

**Comment:**

The paper proposes to use LLM to generate environments for training embodied agents.  Based on the agent's performance, new environments are iteratively generated to focus on training agents on the skills where they are weak.  Experiments on Crafter and Heist show that the proposed method can lead to agents that performs better than baselines.

Reviewers are in general positive on the submission, noting that the paper proposes a relatively novel use of LLMs for training agents.  Reviewers also noted the paper is well written (oNhz,y9sf) with experiments that support the usefulness of the proposed idea (oNhz).  However, reviewers had questions about the applicability of the method to more complex environments (Nrar), for what environments and why are LLMs helpful (oNhz,y9sf), and lack of discussion of relation to curriculum learning (oNhz).  Despite the weaknesses noted by the reviewers, reviewers remain positive on the work.  The AC agrees the work presents a novel and interesting perspective on the use of LLMs for training agents and thus recommends acceptances.

The AC encourages the authors to incorporate feedback from the reviewers and clarifications from the rebuttal for the camera ready, including:
- Additional experiments comparing against curriculum learning (oNhz, y9sf)
- Improve discussion of relation to curriculum learning and limitations of using LLMs to generate environments (what knowledge is leveraged, handling of complex environments, does generated environments need to match 'real' environments, etc)
- Address other points of clarifications noted by reviewers